# Clinical Outcomes and Cost Analysis in Patients with Heart Failure Undergoing Transcatheter Edge-to-Edge Repair for Mitral Valve Regurgitation

**DOI:** 10.3390/jpm14090978

**Published:** 2024-09-15

**Authors:** Aleksander Dokollari, Serge Sicouri, Roberto Rodriguez, Eric Gnall, Paul Coady, Farah Mahmud, Stephanie Kjelstrom, Georgia Montone, Yoshiyuki Yamashita, Jarrett Harish, Beatrice Bacchi, Rakesh C Arora, Ashish Shah, Nitin Ghorpade, Sandra Abramson, Katie Hawthorne, Scott Goldman, William Gray, Francesco Cabrucci, Massimo Bonacchi, Basel Ramlawi

**Affiliations:** 1Department of Cardiac Surgery Research, Lankenau Institute for Medical Research, Main Line Health, Wynnewood, PA 19096, USA; sicouris@mlhs.org (S.S.); ramlawib@mlhs.org (B.R.);; 2Cardiac Surgery Division, St. Boniface Hospital, University of Manitoba, Winnipeg, MB R2H 2A6, Canada; 3Department of Cardiac Surgery, Lankenau Heart Institute, Main Line Health, Wynnewood, PA 19096, USA; 4Division of Cardiology, Lankenau Heart Institute, Main Line Health, Wynnewood, PA 19096, USA; 5Cardiac Surgery F.U., Experimental and Clinical Medicine Department, University of Florence, 50134 Firenze, Italymassimo.bonacchi@unifi.it (M.B.); 6Division of Cardiac Surgery, Harrington Heart and Vascular Institute, University Hospitals, Cleveland, OH 44106, USA

**Keywords:** TEER, echocardiographic outcomes, heart failure

## Abstract

*Objective:* To analyze the clinical and cost outcomes of transcatheter edge-to-edge repair (TEER) for mitral regurgitation (MR) in heart failure (HF) patients. *Methods:* All 162 HF patients undergoing TEER for MR between January 2019 and March 2023 were included. A propensity-adjusted analysis was used to compare 32 systolic vs. 97 diastolic vs. 33 mixed (systolic + diastolic) HF patients. Systolic, diastolic, and mixed HF patients were defined according to AHA guidelines. The primary outcome was the long-term incidence of all-cause death and major adverse cardiovascular and cerebrovascular events (MACCEs, all-cause mortality + stroke + myocardial infarction + repeat intervention). *Results:* The mean age was 76.3 vs. 80.9 vs. 76 years old, and the mean ejection fraction (EF) was 39.5% vs. 59.8% vs. 39.7% in systolic vs. diastolic vs. mixed HF, respectively. Postoperatively, the diastolic vs. systolic HF group had a higher intensive care unit stay (21 vs. 0 h; HR 67.5 (23.7, 111.4)]; lower ventilation time [2 vs. 2.3 h; HR 49.4 (8.6, 90.2)]; lower EF [38% vs. 58.5%; HR 9.9 (3.7, 16.1)]. In addition, the diastolic vs. mixed HF groups had a lower incidence of EF < 50% (11 vs. 27 patients; HR 6.6 (1.6, 27.3) and a lower use of dialysis (one vs. three patients; HR 18.1 (1.1, 287.3), respectively. At a mean 1.6 years follow-up, all-cause death [HR 39.8 (26.2, 60.5)], MACCEs [HR 50.3 (33.7–75.1)], and new pacemaker implantations [HR 17.3 (8.7, 34.6)] were higher in the mixed group. There was no significant total hospital cost difference among the systolic (USD 106,859) vs. diastolic (USD 91,731) vs. mixed (USD 120,522) HF groups (*p* = 0.08). *Conclusions:* TEER for MR evidenced the worst postoperative and follow-up clinical outcomes in the mixed HF group compared to diastolic and systolic HF groups. No total hospital cost differences were observed.

## 1. Introduction

Transcatheter edge-to-edge repair (TEER) in patients with mitral valve regurgitation (MR) has proven to be an effective treatment for patients not eligible for surgical repair [1]. The benefits of TEER for MR include a fast recovery time and high procedural safety [2]. Although several studies have compared TEER outcomes in patients with MR and heart failure (HF) [2,3,4], most of them reported only 30-day outcomes and lacked granularity on postoperative echocardiographic variables. Importantly, most of these studies combined the outcomes of systolic (HFpEF) and diastolic (HFrEF) HF patients. In addition, the clinical outcomes of patients with preserved (HFpEF) vs. reduced ejection fraction (HFrEF) after TEER for MR with the Mitraclip were not reported. Moreover, the impact of postoperative medical therapy in patients with HF on long-term prognosis has also not been adequately investigated. A randomized clinical trial has proven that TEER for MR has better outcomes when compared to conservative medical management [5]. The COAPT (Cardiovascular Outcomes Assessment of the MitraClip Percutaneous Therapy for Heart Failure Patients With Functional Mitral Regurgitation) randomized clinical trial showed a significant reduction in HF and all-cause death at 2 years follow-up in patients undergoing TEER for MR when compared to conservative medical management [6]. On the other hand, negative results produced by the MITRA-FR (Multicentre Study of Percutaneous Mitral Valve Repair MitraClip Device in Patients with Severe Secondary Mitral Regurgitation) trial have been explained by the inclusion of patients at more advanced stages of LV disease but with mild MR [7]. Therefore, several issues are apparent with the interpretation of clinical outcomes from the randomized COAPT and MITRA-FR clinical trials. The goal of this study is to analyze clinical and echocardiographic outcomes as well as hospital costs in patients undergoing TEER for MR with the MitraClip device for systolic vs. diastolic vs. combined (systolic + diastolic, HFmrEF) HF. In addition, a secondary analysis of patients with primary MR was conducted.

## 2. Materials and Methods

### 2.1. Study Population

We identified all patients with HF who underwent TEER for MR between March 2017 and September 2022 at the Lankenau Heart Institute (Lankenau Medical Center, Wynnewood, PA, USA). The study protocol was approved by the Main Line Health Hospitals Institutional Review Board (IRB 45CFR164.512). Patients’ individual consent was waived due to the retrospective nature of the study. All consecutive patients who underwent isolated TEER for MR nonresponsive to medical treatment and deemed at high surgical risk were included in this study. All patients were considered suitable for TEER repair with the MitraClip device based on preprocedural transesophageal echocardiography. Patients were identified via operation codes in a digital operation registry database for all TEER operations.

### 2.2. Patients’ Follow-Up

Follow-up was performed at our outpatient clinic and from the hospital registry. All patients had at least one follow-up time point available. In case the patient did not show up to a follow-up visit, we called the referring cardiologist to acquire the information for this study.

### 2.3. Primary and Secondary Goals and Definitions

The primary outcome was the long-term incidence of all-cause death and major adverse cardiovascular and cerebrovascular events (MACCEs, all-cause mortality + stroke + myocardial infarction + repeat intervention, whether transcatheter or through open-heart surgery). MR was diagnosed based on the patient’s preoperative transthoracic (TTE) and transesophageal echocardiography (TEE) findings, and was graded according to current guidelines based on 2-dimensional Doppler echocardiography; the severity of MR was quantified using four semiquantitative grades as described by the American Society of Echocardiography (grade 0 indicating none, 1+ mild, 2+ moderate, and 3+ severe).

#### Definition of Systolic vs. Diastolic vs. Mixed HF

HF and all variables and outcomes were defined according to the American Heart Association clinical guidelines [8].

### 2.4. Patients’ Variables

Variables collected for each patient included age, gender, race, STS-PROM risk of mortality, body mass index (BMI), obesity, creatinine level, comorbidities such as preoperative dialysis, smoking, chronic obstructive pulmonary disease (COPD), hypertension, dyslipidemia, cerebrovascular disease (CBVD), peripheral vascular disease (PVD), liver disease, diabetes, mediastinal radiation, prior PCI, prior CABG, prior MI, prior valve surgery, Afib, ejection fraction (EF), number of diseased vessels, left main coronary artery stenosis, severe proximal LAD lesion, LITA, and radial artery graft use.

### 2.5. Statistical Analysis

Descriptive statistics were used for all pre-, intra-, and postoperative variables. Initial comparison of pre-, intra-, postoperative, and echocardiographic variables by heart failure groups were performed with one-way ANOVA (parametric) or Kruskal–Wallis (non-parametric) tests for continuous variables and chi-square or Fisher’s tests for categorical variables. Continuous variables were assessed with histograms and the rule of central tendency to determine normality, and displayed as means (standard deviations) for normally distributed variables or medians (interquartile range) for non-normally distributed variables. Three propensity scores were created using multivariable logistic regression with the systolic HF vs. diastolic HF, systolic HF vs. mixed HF, and diastolic vs. mixed HF groups as the dependent variables for each score. Preoperative variables that differed between the groups were entered into the models as independent variables. Intra- and postoperative outcomes were compared with propensity score-adjusted regression models with beta coefficients displayed for continuous variables, odds ratios for categorical variables, and 95% confidence intervals. See the Appendix A for additional explanations of the propensity score and model building.

The long-term outcomes of the HF groups were compared with the cumulative incidence per 100 person-years and log-rank tests; the number of risks and frequency and the percentage of events at 1, 2, and 5 years; finally, univariable, propensity-adjusted, and multivariable Cox regression and Fines and Grays analyses. After fitting these survival models, we created survival and cumulative incidence graphs using the adjusted results and the curve function in Stata. In addition, cubic spline graphs using a GLM with Poisson family, log link, and robust estimates were produced to visualize changes in the risk ratio over time. Finally, risk factors for the long-term outcomes of the HF group were found by using a forward selection with high *p*-value criteria (0.5) and AICs. The risk factor analysis used Cox regression for all-cause mortality, and MACCEs and Fines and Grays competing risk regression for all other outcomes. All analyses were performed in Stata 17.0 (StataCorp, LLC. College Station, TX, USA). The 95% confidence intervals and *p*-values are reported, with a *p*-value < 0.05 considered significant.

### 2.6. Propensity Score Adjustment Significance Compared to Propensity Score Matching

Propensity matching provides excellent matching before the analysis of the preoperative data, while the propensity adjustment accounts for biases during the analysis and adjusts the intraoperative and postoperative data. Therefore, while seeing significant differences between preoperative variables, these differences are adjusted during the modeling process. Propensity matching reduces the size of the groups while propensity adjustment retains the sample size of the groups. This method is particularly suitable for smaller sample sizes [9]. In addition, the propensity adjustment process correctly adjusts the differences among populations, regardless of the fact that one of the groups presented with a lower EF or had a higher Society of Thoracic Surgeons (STS) score. In this context, a problem with propensity matching is the allocation bias, which is correctly adjusted in the propensity adjustment analysis, therefore providing homogeneous outcomes when comparing different populations.

### 2.7. Cost Analysis

We performed a cost analysis between the systolic vs. diastolic vs. mixed HF groups. The total hospital cost for each year was divided into direct + indirect costs. Direct cost expenses included surgical OR time, hospital stay, surgical implants and supplies, catheterization laboratory, repeat intervention, pharmacy, radiology and ultrasound imaging, blood bank, cardiology, emergency department, physical medicine, laboratory, respiratory therapy, hospital readmission, and physician fees. Indirect cost expenses included general administration, medical records, information technology, physical plant maintenance, human resources, volunteer services, finance, and other regional services.

Using the propensity-adjusted groups, we performed a cost analysis for each patient from January 2018 to September 2022, which was the date frame of available cost data due to system records. We did not adjust for inflation.

## 3. Results

### 3.1. Systolic vs. Diastolic vs. Combined HF Preoperative Characteristics

A total of 32 (systolic HF) vs. 97 (diastolic HF) vs. 33 (combined HF) patients were included in the analysis, respectively (Table 1). The mean age was 76.3 vs. 80.9 vs. 76 years old, the mean STS risk score was 5% in all groups, and the mean EF was 39.5% vs. 59.8% vs. 39.7% in the systolic vs. diastolic vs. mixed HF groups, respectively. A total of 15 (systolic HF), 84 (diastolic HF), and 26 (combined HF) with a primary MR mechanism were part of the analysis, while the remainder of the patients had secondary MR.

### 3.2. Intraoperative Outcomes

Intraoperatively, there were no significant differences among the groups, while there was only one conversion to full sternotomy in the diastolic group due to acute decompensation of the patient due to hemodynamic instability (Table 2). In addition, almost 80% of the patients were extubated in the operating room.

### 3.3. Postoperative Outcomes

Postoperatively, after the propensity adjustment, diastolic HF patients had higher total ventilation hours [OR 49.4 (8.6, 90.2)], intensive care unit (ICU) length of stay [OR 67.5 (23.7, 111.4], a lower mean EF [HR 9.9 (3.7, 16.1)], and higher warfarin use [HR 10.8 (1.5, 75.7)] compared to systolic HF patients. In addition, diastolic HF patients had a higher mean EF compared to the other two groups. On the other hand, the combined HF group had a higher incidence of EF < 50% [OR 6.6 (1.6, 27.3)] and dialysis [OR 18.1 (1.1, 287.3)] compared to the diastolic HF group. In addition, the combined HF group had a higher left ventricular end-systolic and end-diastolic diameter compared to the systolic and diastolic HF groups (Table 3, Table 4 and Table 5). The sub-analysis of only the propensity-adjusted primary MR showed that the diastolic HF group had a higher ICU stay [R 32.7 (7.9, 57.5)] and EF < 50% 7.1 (2.0, 12.1) vs. the systolic HF group. In addition, patients with combined HF had a higher incidence of EF < 50% [OR 9.5 (1.9, 47.0)] and dialysis [OR 9.5 (1.9, 47.0)] vs. diastolic HF patients.

### 3.4. Patients’ Follow-Up

At the 5-year follow-up, all-cause mortality [HR 39.8 (26.2, 60.5); *p* = 0.002], MACCEs (all-cause mortality + stroke + myocardial infarction [(MI) + repeat intervention) (HR 50.3 (33.7–75.1); *p* = 0.002], and new pacemaker implantations [HR 17.3 (8.7, 34.6); *p* = 0.001] were higher in the combined HF group vs. the systolic and diastolic HF groups (Table 3, Figure 1, Figure 2 and Figure 3). A sensitivity analysis at the 1-, 2-, and 5-year follow-up and a multivariate Cox regression analysis evidenced a significantly higher incidence of all-cause mortality, MACCEs, and new pacemaker implantations in the combined HF group (Table 4 and Appendix A). Echocardiographic outcomes at follow-up evidenced that the diastolic HF group had a higher mean EF and a lower left ventricular end-systolic diameter. Risk predictors impacting the long-term outcomes in the systolic HF group included dialysis, previous mediastinal radiation, previous CI, prior MI, and being immunocompromised. Risk predictors impacting the long-term outcomes in the diastolic HF group included New York Heart Association (NYHA) class ≥ 2, being immunocompromised, prior valve surgery, age, cardiogenic shock, and home therapy O_2_ use. Risk predictors impacting the long-term outcomes in the combined HF group included pneumonia, transitory ischemic attack, age, diabetes, prior valve surgery, and chronic obstructive pulmonary disease (Table 6A–C). The follow-up completion rate was 97%.

### 3.5. Cost Analysis

The cost analysis in the adjusted cohort evidenced that the total hospital costs were not significantly different between the systolic (USD 106,859 × patient) vs. diastolic (USD 91,731 × patient) vs. mixed (USD 120,522 × patient) HF groups (*p* = 0.08) (Table 7). Direct costs among the systolic (USD 65,362 × patient) vs. diastolic (USD 57,094 × patient) vs. mixed (USD 72,465 × patient) HF groups (*p* = 0.118) were not significantly different. Indirect costs among the systolic (USD 41,496 × patient) vs. diastolic (USD 34,637 × patient) vs. mixed (USD 48,058 × patient) HF groups (*p* = 0.04) were lower in the diastolic HF group.

## 4. Discussion

### 4.1. Novelties in Medical Literature

This analysis provided several novel insights into the population of patients with HF and MR undergoing TEER with the MitraClip.
All-cause death, MACCEs, and new pacemaker implantations in patients with combined HF were higher than in patients with diastolic HF.Propensity-adjusted postoperative and echocardiographic outcomes evidenced the worst outcomes for diastolic HF patients compared to systolic HF patients in the entire cohort and primary MR analysis.There was no significant difference in the total hospital cost among the groups.

### 4.2. Comparison with MITRA-FR Clinical Trial

This analysis is in line with several other publications and meta-analyses in the medical literature comparing these three HF groups [10,11]. In this context, the MITRA-FR clinical trial evidenced that, in patients with secondary MR, the rate of repeat hospitalization among patients undergoing TEER and those treated with medical therapy did not differ [7]. However, the trial failed to provide granular data on the outcomes based on the specific type of heart failure. This is important because the underlying pathophysiology of systolic vs. diastolic vs. combined HF is radically different. In this context, left ventricular volumes and diameters are drastically different among groups, and this has been shown to play an important role in clinical outcomes [12]. However, the clinical and physiological evolution of systolic HF progresses to diastolic HF if left untreated. This study evidenced that patients with either diastolic HF or combined HF have worse postoperative and long-term outcomes when compared to patients with systolic HF, and that this is observed in primary as well as secondary MR. In this context, the MITRA-FR randomized clinical trial showed an all-cause death of 24.3% in patients with HF at only 1-year follow-up [7]. Our study reported an overall all-cause death at 1 year of 13.4–30.3%, at 2 years of 18.6–51.5%, and at 5 years of 32–66.7%. As noticed, there is a disparity among the mean of all-cause mortality among the MITRA-FR trial and our diastolic HF (diastolic HF = 13.4% vs. MITRA-FR = 24.3%) and patients with combined HF (combined HF = 24.3% vs. MITRA-FR = 30.3%) groups. Therefore, it is important to understand that the type of HF can make a difference in clinical outcomes. However, it must be stated that all of the patients in the MITRA-FR trial had secondary MR, while more than half of our population had primary MR.

### 4.3. Risk Predictors Impacting Clinical Outcomes

Another important addition to the medical literature from this study is the presence of risk predictors impacting clinical outcomes [13]. This is important because there are modifiable risk factors that can be worked on to reduce the incidence of all-cause mortality, MACCEs, stroke, MI, and pacemaker implantations after TEER for MR in patients with HF. In this context, our study reported that four clinical variables (the NYHA functional class, chronic obstructive pulmonary disease, atrial fibrillation or flutter, and chronic kidney disease) and four echocardiographic variables (left ventricular ejection fraction, left ventricular end-systolic dimension, right ventricular systolic pressure, and tricuspid regurgitation) are predictors for all-cause mortality in the COAPT clinical trial [14]. However, the present study added risk predictors, not only for all-cause mortality, but also for cardiac death, MACCEs, stroke, MI, repeat interventions, and new pacemaker implantations based on the type of HF (systolic vs. diastolic vs. combined). This is relevant because the quality of life after an episode of stroke can be miserable, and being able to have adequate prevention is crucial for the success of its long-term prognosis.

### 4.4. Evolution of HF

In patients with early HF with reduced EF (HFrEF), left ventricular chamber dilatation increases mitral valve annulus dimensions and papillary muscle displacement (and possibly dysfunction), leading to the functional alteration of the mitral valve apparatus. Consequently, the left ventricle undergoes an adaptive dilatative remodeling in order to accommodate the volume overload due to valvular regurgitation. As a result, myocardial contraction force increases as preload increments in accordance with the Frank–Starling law. This phenomenon allows for the maintenance of an appropriate emptying of the enlarged left ventricle and the preservation of the stroke volume and cardiac output. With the progression of the disease and the increment of volume overload, patients can develop atrial fibrillation secondary to left atrium dilatation and remodeling, while left ventricular and annular dilation increase. All of this leads to increased leaflet tethering and decreased closing forces, perpetuating the vicious cycle. The MitraClip treatment of patients with advanced HF (as with most patients from the MITRA-FR trial) may not be effective. Similarly, in HFpEF, if the TEER is performed late, the benefits may be limited by the presence of advanced diastolic dysfunction and comorbidities. Performing MitraClip earlier in the disease process might yield better outcomes in both groups, though this is often more difficult to achieve in HFpEF due to diagnostic and referral delays. As a matter of fact, patients with “early HF” (mild left ventricular dilatation, and no or few hospitalizations for HF) and severe MR (EROA > 40 mm^2^) are more likely to be the best patients to be treated with the MitraClip therapy, as also observed in the COAPT trial. Our observations are in line with the recently reported concept of the EROA/LVEDV ratio as a marker of “disproportionate” or “proportionate” MR [15,16].

Another aspect to consider is that, while the MitraClip can reduce MR, the long-term benefit might be limited if diastolic dysfunction and comorbidities dominate the clinical picture. Moreover, pulmonary hypertension, which is often more severe in HFpEF, can complicate this scenario. Even after successful MitraClip therapy, elevated pulmonary pressures may persist, limiting the improvement in symptoms and survival in HFpEF patients [17]. In these patients, while the MitraClip can reduce symptoms by decreasing left atrial pressure, the underlying diastolic dysfunction remains, which might limit the overall clinical improvement. HFpEF patients, despite a normal EF, may have a limited ability to increase cardiac output due to stiff ventricles, meaning that MR reduction may not translate into significant clinical improvement. In addition, the stiff, non-compliant left ventricle in HFpEF may not respond as favorably to the MitraClip therapy, since the primary issue is not volume overload but rather filling pressures [18].

Furthermore, in both HFrEF and HFpEF, chronic MR leads to left atrial enlargement and dysfunction. However, HFpEF patients may have more significant atrial fibrosis and dysfunction, which can persist even after MR is treated, limiting the overall benefit of the MitraClip [19,20].

### 4.5. Procedural Factors

HFpEF patients might present anatomical challenges, such as calcified valves or small left ventricular cavities, making the MitraClip procedure technically more difficult and potentially less effective, and with a longer intraoperative time [21].

In contrast, HFrEF patients, despite often having dilated ventricles, might have more favorable anatomy for the procedure. The incomplete reduction in MR during the procedure, more common in complex cases, can lead to suboptimal outcomes. Residual MR is more likely in patients with HFpEF due to structural complexities [22].

### 4.6. Cost Analysis

The medical literature has several clinical studies describing the cost benefits of TEER for MR with the Mitraclip [23]. However, our study is the first one to specifically analyze the cost analysis in patients with HFrEF vs. HFpEF vs. HFmrEF. The data show higher costs when compared to the COAPT clinical trial. However, patients in the COAPT clinical trial had secondary MR only. In our study, with cost not being a differentiating factor, clinicians can focus more on patient outcomes, safety, and satisfaction when choosing treatment options. Clinicians may find it easier to make treatment decisions without needing to consider cost variations, simplifying the clinical decision-making process. In terms of healthcare policy, if cost is not a differentiating factor, clinicians and healthcare administrators may need to reassess how they measure cost-effectiveness. This could lead to greater emphasis on metrics like quality-adjusted life years (QALYs) or patient satisfaction rather than purely financial metrics. In addition, policymakers could advocate for more equitable access to treatments. Lastly, policies could evolve to support more personalized and patient-centered care models, as financial constraints would no longer be a barrier to offering tailored treatments. This may encourage the healthcare system to shift toward more value-based care that prioritizes individual patient needs. The lack of significant cost differences should encourage both clinical practice and healthcare policy to focus on patient outcomes, equitable access, and standardized care while simplifying administrative processes and supporting healthcare innovations. This shift could ultimately lead to better patient experiences and more sustainable healthcare systems.

## 5. Limitations

This retrospective study was subject to all limitations inherent to a nonrandomized study, including the completeness of the follow-up and the potential selection bias including patients who underwent the Mitraclip procedure. The rigorous propensity-matched analysis, however, limited these biases. Despite our best efforts to exclude confounding factors through multivariate analysis, the potential for unknown confounders still exists. Another limitation is the single-center data, as institutional practices and patient demographics, as well as hospital-based charges, could differ from other settings. Therefore, this analysis warrants further validation from multicenter studies.

## 6. Conclusions

TEER for MR evidenced the worst postoperative and follow-up outcomes in mixed HF patients compared to diastolic and systolic HF patients. There were no significant hospital cost differences among the groups. Understanding the best timing for intervention with TEER in patients with HF is crucial for good long-term clinical outcomes.

## Figures and Tables

**Figure 1 jpm-14-00978-f001:**
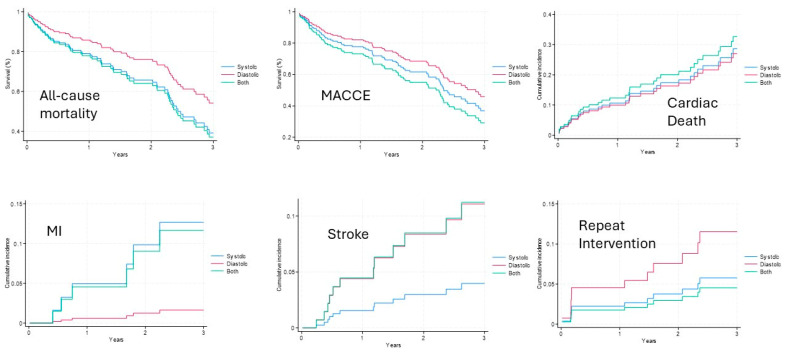
Propensity-adjusted survival and cumulative incidence curves. Legend: All-cause mortality—MACCEs—Cardiac Death—Myocardial Infarction—Stroke—Repeat intervention Curve.

**Figure 2 jpm-14-00978-f002:**
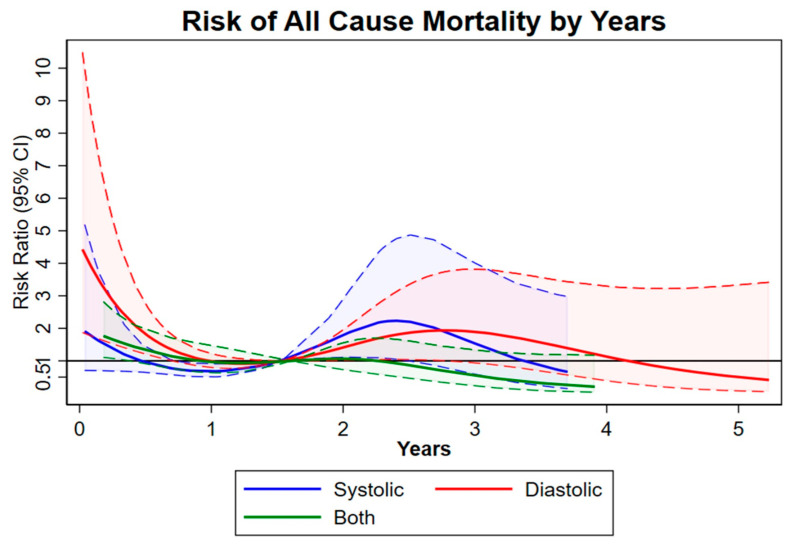
HR for all-cause mortality. Dash-lines are deviations from the mean.

**Figure 3 jpm-14-00978-f003:**
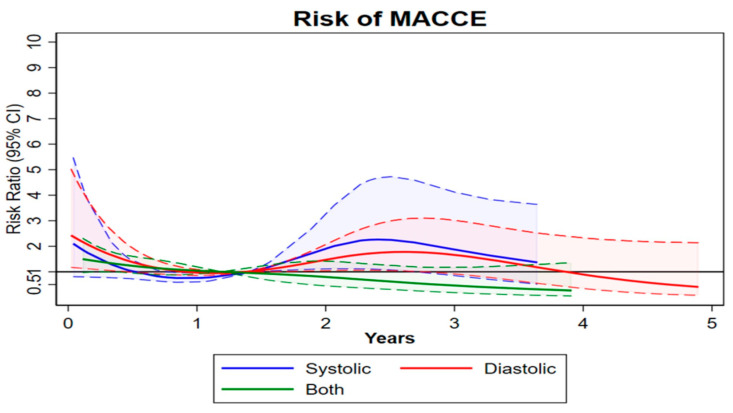
HR for MACCEs; stroke; MI; repeat intervention. Dash-lines are deviations from the mean.

**Table 1 jpm-14-00978-t001:** Preoperative characteristics.

Preoperative Variables	Systolic *n* = 32	Diastolic *n* = 97	Mixed *n* = 33	*p*-Value
Age (mean/SD)	76.3 (7.6)	80.9 (9.2)	76.0 (8.9)	**0.003**
Gender, *n* (%)				**0.014**
Male	20 (62.5%)	37 (38.1%)	20 (60.6%)	
Race, *n* (%)				**0.02**
White	32 (100%)	87 (89.7%)	26 (78.8%)	
Other	0 (0)	10 (10.3%)	7 (21.1%)	
NYHA functional classification, *n* (%)				0.070
Class I or II	15 (46.9%)	45 (46.4%)	8 (24.4%)	
Class III or IV	17 (53.1%)	52 (53.6%)	25 (75.8%)	
STS-PROM risk of mortality (median/IQR)	0.05 (0.03–0.1)	0.05 (0.03–0.08)	0.05 (0.03–0.1)	0.28
BMI kg/m^2^ (Mean/SD)	26.0 (4.6)	24.7 (4.8)	26.6 (6.1)	0.125
Obese, *n* (%)	6 (18.8%)	16 (16.5%)	8 (24.2%)	0.612
Creatinine level (Median/IQR)	1.7 (1.2–2)	1 (0.8–1.3)	1.4 (1.1–2.1)	**0.0001**
Albumin level (Median/IQR)	3.6 (3.2–4)	3.7 (3.4–3.9)	3.6 (2.9–3.8)	0.136
proBNP level pg/mL (Median/IQR)	543 (312–1206)	380 (218–666)	943 (437–2064)	**0.0001**
Dialysis, *n* (%)	2 (6.3%)	0 (0)	5 (15.2%)	**0.001**
Smoking, *n* (%)	17 (53.1%)	45 (46.4%)	18 (54.6%)	0.645
COPD, *n* (%)	9 (28.1%)	21 (21.7%)	5 (15.2%)	0.446
CKD, *n* (%)	25 (78.1%)	43 (44.3%)	25 (75.8%)	**<0.0001**
Pneumonia, *n* (%)	6 (18.8%)	12 (12.4%)	5 (15.2%)	0.659
Home O2, *n* (%)	2 (6.3%)	11 (11.3%)	4 (12.1%)	0.677
Hypertension, *n* (%)	28 (87.5%)	74 (76.3%)	29 (87.9%)	0.195
Dyslipidemia, *n* (%)	28 (87.5%)	61 (62.9%)	19 (57.6%)	**0.017**
CBVD, *n* (%)	9 (28.1%)	16 (16.5%)	6 (18.2%)	0.345
PVD, *n* (%)	10 (31.3%)	10 (10.3%)	4 (12.1%)	**0.014**
Liver disease, *n* (%)	2 (6.3%)	2 (2.1%)	1 (3.0%)	0.494
Diabetes, *n* (%)	11 (34.3%)	15 (15.5%)	11 (33.3%)	0.024
Immunocompromise, *n* (%)	1 (3.1%)	11 (11.3%)	0 (0)	0.06
Cancer w/in 5 months, *n* (%)	2 (6.3%)	14 (14.4%)	5 (15.2%)	0.448
Prior mediastinal radiation, *n* (%)	4 (12,5%)	9 (9.3%)	1 (3.0%)	0.374
Prior cardiovascular intervention, *n* (%)	24 (75.0%)	42 (43.3%)	19 (57.6%)	**0.006**
Previous PCI, *n* (%)	16 (50.0%)	18 (18.6%)	11 (33.3%)	**0.002**
Prior CABG, *n* (%)	12 (37.5%)	20 (20.6%)	8 (24.2%)	0.158
Prior valve surgery, *n* (%)	6 (18.8%)	9 (9.3%)	7 (21.2%)	0.143
Prior MI, *n* (%)	17 (53.1%)	21 (21.7%)	13 (39.4%)	**0.002**
Prior stroke, *n* (%)	5 (15.6%)	5 (5.2%)	3 (9.1%)	0.162
Prior TIA, *n* (%)	3 (9.4%)	14 (14.4%)	4 (12.1%)	0.751
Mitral valve stenosis, *n* (%)	3 (9.4%)	8 (8.3%)	0 (0)	0.216
Cardiogenic shock, *n* (%)	4 (12.5%)	4 (4.1%)	6 (18.2%)	**0.032**
History of atrial fibrillation, *n* (%)	24 (75.0%)	64 (66.0%)	28 (84.9%)	0.103
Prior history of arrhythmias, *n* (%)	28 (87.5%)	63 (65.0%)	28 (84.9%)	**0.011**
Previous AICD, *n* (%)	8 (25.0%)	3 (3.1%)	6 (18.2%)	**0.001**
Previous pacemaker, *n* (%)	8 (25.0%)	11 (11.3%)	8 (24.2%)	0.084
Mitral regurgitation primary mechanism, *n* (%)				**<0.0001**
Functional mitral regurgitation	17 (53.1%)	13 (13.4%)	7 (21.1%)	
Degenerative mitral regurgitation	10 (31.3%)	82 (84.5%)	24 (72.7%)	
Mixed mitral regurgitation	5 (15.6%)	2 (2.1%)	2 (6.1%)	
EF (mean/SD)	39.5 (15.0)	59.8 (11.5)	39.7 (14.7)	**<0.0001**
EF < 50% *n* (%)	22 (68.8%)	9 (9.3%)	25 (75.8%)	**<0.0001**
Creatinine clearance (mean/SD)	42.4 (22.7)	48.2 (22.9)	42.9 (17.7)	0.261
Creatinine clearance < 60 mL/min *n* (%)	28 (87.5%)	70 (72.2%)	25 (75.8%)	0.213
Coronary diseased vessels, *n* (%)				**0.044**
0	7 (21.9%)	46 (47.4%)	8 (24.2%)	
1	5 (15.6%)	18 (18.6%)	9 (27.3%)	
2	8 (25.0%)	15 (15.5%)	10 (30.3%)	
3	12 (37.5%)	17 (17.5%)	6 (18.2%)	
4	0 (0)	1 (1.0%)	0 (0)	

Note: AICD: automatic implantable cardioverter defibrillator.

**Table 2 jpm-14-00978-t002:** Intraoperative and postoperative outcomes.

Variables	Systolic *n* = 32	Diastolic *n* = 97	Mixed *n* = 33	*p*-Value	PS Adjusted Analysis		
					Diastolic vs. Systolic	Systolic vs. Mixed	Diastolic vs. Mixed
Intraoperative Outcomes					β or OR (95% CI)	β or OR (95% CI)	β or OR (95% CI)
Time in OR (hours) (mean/SD)	2.8 (0.7)	2.9 (0.6)	2.7 (0.6)	0.349	0.1 (−0.2, 0.5)	−0.1 (−0.5, 0.3)	−0.1 (−0.5, 0.2)
All type blood transfusion, *n* (%)	0 (0)	3 (3.1%)	0 (0)	0.359	NA	NA	NA
Extubated in OR, *n* (%)	30 (85.7%)	84 (78.5%)	21 (61.8%)	0.049	0.3 (0.1, 1.3)	0.2 (0.04, 0.9)	0.9 (0.3, 3.1)
Conversion to sternotomy	0 (0)	1 (1.0%)	0 (0)	0.714	NA	NA	NA
Surgery priority, *n* (%)				0.124			
Elective, *n* (%)	28 (80%)	90 (84.1%)	22 (64.7%)		Ref	Ref	Ref
Urgent/emergent, *n* (%)	7 (20%)	17 (15.9%)	12 (35.3%)		1.5 (0.4, 6.1)	1.9 (0.5, 8.4)	1.8 (0.5, 6.2)
Clip numbers, *n* (%)				0.754			
0–1	19 (59.4%)	51 (52.6%)	19 (57.6%)		Ref	Ref	Ref
2–4	13 (40.6%)	46 (47.4%)	14 (42.4%)		1.3 (0.4, 3.8)	0.8 (0.2, 2.9)	0.9 (0.3, 2.9)
Postoperative Outcomes					β or OR (95% CI)	β or OR (95% CI)	β or OR (95% CI)
Total ICU (hours) (median/IQR)	21 (0–36.2)	0 (0–29)	0 (0–104)	0.461	67.5 (23.7, 111.4)	29.7 (−64.1, 123.5)	−28.9 (−96.5, 38.7)
Total hospital LOS (days) (Median/IQR)	1 (1–8.5)	1 (1–3)	3 (1–16)	0.046	2.7 (−0.4, 5.8)	1.7 (−5.1, 8.4)	1.1 (−3.2, 5.4)
Total ventilation hours (median/IQR)	2 (2–2.7)	2.3 (2–3)	2.2 (2–6)	0.08	49.4 (8.6, 90.2)	17.0 (−38.9, 72.9)	−48.5 (−99.5, 2.5)
Prolonged ventilation > 24 h, *n* (%)	1 (3.1%)	4 (4.1%)	4 (12.1%)	0.178	9.1 (0.6, 148.5)	1.2 (0.1, 18.2)	0.6 (0.1, 4.7)
RBC units, *n* (%)	5 (15.6%)	7 (7.2%)	8 (24.2%)	0.03	1.3 (0.2, 7.5)	1.5 (0.3, 7.9)	1.8 (0.4, 9.0)
Cryoprecipitate units, *n* (%)	0 (0)	0 (0)	1 (3.0%)	0.14	NA	NA	NA
Platelet units, *n* (%)	0 (0)	0 (0)	1 (3.0%)	0.14	NA	NA	NA
FFP units *n* (%)	0 (0)	0 (0)	0 (0)	0.999	NA	NA	NA
EF (mean/SD)	38.0 (15.8)	58.5 (10.9)	39.4 (13.8)	<0.0001	9.9 (3.7, 16.1)	−1.3 (−11.1, 8.4)	−1.8 (−6.7, 3.1)
EF < 50% *n* (%)	22 (68.8%)	11 (11.3%)	27 (81.8%)	<0.0001	0.2 (0.1, 0.5)	3.0 (0.6, 14.2)	6.6 (1.6, 27.3)
Creatinine level (mean/SD)	1.7 (0.8)	1.2 (0.9)	2.0 (1.4)	0.0014	0.3 (−0.1, 0.8)	0.2 (−0.6, 0.9)	−0.3 (−0.9, 0.2)
Creatinine clearance (mean/SD)	45.6 (27.3)	47.9 (23.2)	41.1 (18.6)	0.351	−0.7 (−14.1, 12.6)	−6.9 (−22.4, 8.4)	0.3 (−12.0, 12.6)
Creatinine clearance < 60 mL/min, *n* (%)	27 (84.4%)	71 (73.2%)	28 (84.9%)	0.23	0.4 (0.1, 1.8)	1.8 (0.3, 10.9)	1.7 (0.4, 6.9)
Stroke, *n* (%)	0 (0)	2 (2.1%)	0 (0)	0.507	NA	NA	NA
CVA/TIA, *n* (%)	0 (0)	3 (3.1%)	0 (0)	0.359	NA	NA	NA
Dialysis, *n* (%)	2 (6.3%)	1 (1.0%)	3 (9.1%)	0.074	0.5 (0.02, 12.3)	1.4 (0.1, 15.7)	18.1 (1.1, 287.3)
MI, *n* (%)	0 (0)	1 (1.0%)	0 (0)	0.714	NA	NA	NA
Cardiac arrest, *n* (%)	1 (3.1%)	0 (0)	1 (3.0%)	0.221	NA	0.3 (0.01, 9.6)	NA
Endocarditis, *n* (%)	0 (0)	0 (0)	0 (0)	0.999	NA	NA	NA
Postoperative atrial fibrillation *n* (%)	3 (9.4%)	4 (4.1%)	2 (6.1%)	0.526	0.2 (0.02, 1.3)	0.2 (0.02, 2.4)	0.99 (0.1, 11.9)
30-day hospital all-cause readmission, *n* (%)	18 (56.3%)	49 (50.5%)	21 (63.6%)	0.413	0.7 (0.2, 2.1)	1.7 (0.4, 6.4)	2.0 (0.6, 6.3)
30-day cardiac readmission	9 (28.1%)	34 (35.1%)	14 (42.2%)	0.482	0.99 (0.3, 3.2)	2.5 (0.6, 10.3)	1.4 (0.4, 4.3)
30-day all-cause mortality, *n* (%)	1 (3.1%)	3 (3.1%)	0 (0)	0.592	1.5 (0.1, 35.2)	NA	NA
Medications, *n* (%)							
Aspirin	25 (78.1%)	76 (79.2%)	25 (75.8%)	0.919	0.7 (0.2, 2.6)	1.1 (0.2, 5.0)	1.4 (0.4, 5.5)
Clopidogrel	15 (46.9%)	46 (47.9%)	13 (39.4%)	0.694	1.3 (0.4, 3.9)	0.5 (0.1, 1.8)	0.9 (0.3, 2.7)
Warfarin	3 (9.4%)	15 (15.6%)	10 (30.3%)	0.065	**10.8 (1.5, 75.7)**	5.3 (0.9, 32.8)	1.5 (0.4, 5.5)
Apixaban	18 (56.3%)	24 (25.0%)	9 (27.3%)	0.004	**0.1 (0.02, 0.4)**	0.3 (0.1, 1.3)	2.0 (0.6, 7.3)
Rivaroxaban	1 (3.1%)	13 (13.5%)	5 (15.2%)	0.229	2.3 (0.2, 26.4)	1.8 (0.2, 4.5)	0.9 (0.2, 4.5)
B-blockers	32 (100%)	97 (100%)	33 (100%)	1	NA	NA	NA
ACE inhibitors	25 (78.1%)	86 (88.6%)	27 (81.8%)	0.641	2.1 (0.3, 22.4)	1.5 (0.3, 3.5)	1.1 (0.3, 4.5)

Note: The *p*-values are derived from ANOVA and chi-square tests. The propensity-adjusted results have 95% confidence intervals instead of *p*-values. OR: odds ratio. Time in OR: time in the operative room; ICU: intensive care unit; LOS: length of stay; FFP: fresh frozen plasma. β: beta coefficient. Odds ratios (OR) are used for categorical variables and β coefficients are used for continuous variables.

**Table 3 jpm-14-00978-t003:** Long-term outcomes.

Cumulative Incidence	Systolic *n* = 32	Diastolic *n* = 97	Mixed *n* = 33	*p*-Value
CI per 100 person-years				
All-cause mortality	33.8 (21.4, 53.8)	16.8 (11.8, 23.9)	39.8 (26.2, 60.5)	0.002
Cardiac death	13.2 (6.3, 27.6)	9.2 (5.7, 14.8)	19.9 (11.0, 35.9)	0.163
MACCEs	38.9 (25.1, 60.4)	22.3 (16.3, 30.5)	50.3 (33.7–75.1)	0.002
MI	3.8 (0.9, 15.2)	1.1 (0.3, 4.4)	4.0 (1.0, 15.9)	0.325
Stroke	9.5 (4.0, 22.8)	2.9 (1.2, 6.9)	5.5 (1.8, 17.2)	0.146
Repeat intervention	3.9 (0.97, 15.4)	5.7 (3.1, 10.6)	3.8 (0.9, 15.0)	0.748
New pacemaker implantation	0	1.7 (0.5, 5.2)	17.3 (8.7, 34.6)	<0.0001

Note: CI: cumulative incidence.

**Table 4 jpm-14-00978-t004:** Follow-up sensitivity analysis.

Year Follow-Up	Number of Patients at Risk			Systolic	Diastolic	Combined	*p*-Value
	Systolic	Diastolic	Combined	*n* = 32	*n* = 97	*n* = 33	
All-Cause Mortality							
1 year	22	74	24	6 (18.8%)	13 (13.4%)	10 (30.3%)	0.09
2 years	14	43	13	9 (28.1%)	18 (18.6%)	17 (51.5%)	0.001
5 years	2	3	1	18 (56.3%)	31 (32.0%)	22 (66.7%)	0.001
Cardiac Death							
1 year	22	74	24	4 (12.5%)	8 (8.3%)	5 (15.2%)	0.492
2 years	14	43	13	4 (12.5%)	12 (12.4%)	10 (30.3%)	0.044
5 years	2	3	1	7 (21.9%)	17 (17.5%)	11 (33.3%)	0.162
MACCEs							
1 year	22	72	20	6 (18.8%)	14 (14.4%)	14 (42.4%)	0.003
2 years	14	38	11	9 (28.1%)	24 (24.7%)	20 (60.6%)	0.001
5 years	2	2	1	20 (62.5%)	39 (40.2%)	24 (72.7%)	0.002
Stroke							
1 year	22	71	24	3 (9.4%)	1 (1.0%)	1 (3.0%)	0.061
2 years	14	39	13	4 (12.5%)	4 (4.1%)	2 (6.1%)	0.233
5 years	2	2	1	5 (15.6%)	5 (5.2%)	3 (9.1%)	0.162
MI							
1 year	21	73	22	1 (3.1%)	0 (0)	2 (6.1%)	0.07
2 years	14	41	11	1 (3.1%)	2 (2.1%)	2 (6.1%)	0.518
5 years	2	3	1	2 (6.3%)	2 (2.1%)	2 (6.1%)	0.401
Reoperation							
1 year	22	71	23	0 (0)	5 (5.2%)	2 (6.1%)	0.397
2 years	14	40	13	0 (0)	8 (8.3%)	2 (6.1%)	0.243
5 years	2	3	1	2 (6.3%)	10 (10.3%)	2 (6.1%)	0.653
New Pacemaker Implantation							
1 year	23	73	19	0 (0)	1 (1.0%)	6 (18.2%)	<0.0001
2 years	13	43	10	0 (0)	2 (2.1%)	7 (21.2%)	<0.0001
5 years	2	2	1	0 (0)	3 (3.1%)	8 (24.2%)	<0.0001

Note: analysis of long-term outcomes. The columns to the right display the frequency and percentages of the events at each time point.

**Table 5 jpm-14-00978-t005:** Echocardiographic preoperative, intraoperative, postoperative, and follow-up.

	Systolic *n* = 32	Diastolic *n* = 97	Mixed *n* = 33	*p*-Value
Preoperative Echocardiographic Characteristics				
EF	40.3 (15.2)	59.6 (11.9)	40.4 (15.1)	<0.0001
MV stenosis (yes, *n*%)	3 (9.4%)	8 (8.3%)	0 (0)	0.216
MR grade (*n*%)				0.143
Moderate	6 (18.8%)	9 (9.3%)	7 (21.2%)	
Severe	26 (81.3%)	88 (90.7%)	26 (78.8%)	
Stroke volume	50.0 (15.9)	57.5 (22.3)	60.8 (18.8)	0.167
LVEDD	5.9 (1.3)	5.2 (0.9)	6.0 (1.3)	0.0004
LVESD	4.6 (1.5)	3.6 (1.1)	4.9 (1.6)	<0.001
LVEDV	155.4 (66.4)	96.4 (32.2)	115.1 (39.2)	<0.001
LVESV	88.4 (52.0)	41.4 (25.4)	61.3 (30.5)	<0.001
MV area	4.1 (1.2)	3.9 (1.3)	4.5 (1.5)	0.198
MV annulus	3.9 (0.8)	3.5 (1.1)	3.9 (0.6)	0.179
Mean gradient	2.5 (2.2)	4.9 (12.2)	2.2 (1.5)	0.367
Regurgitant volume	51.3 (22.3)	58.3 (34.5)	56.1 (30.0)	0.708
Regurgitant fraction	41.3 (32.2)	29.8 (22.0)	36.0 (24.8)	0.29
LAESV	93.6 (46.2)	96.1 (50.1)	100.4 (59.2)	0.897
LA dimension	16.1 (13.8)	16.3 (14.3)	18.8 (15.1)	0.742
RVSP	47.5 (18.2)	45.3 (16.1)	48.4 (14.2)	0.647
Tricuspid valve etiology				0.024
None	1 (3.1%)	0 (0)	1 (3.0%)	
Non-functional	1 (3.1%)	0 (0)	1 (3.0%)	
Functional	30 (93.8%)	97 (100%)	31 (93.9%)	
Tricuspid insufficiency				0.277
None	1 (3.1%)	0 (0)	1 (3.1%)	
Trace	3 (9.4%)	10 (10.5%)	2 (6.3%)	
Mild	11 (34.4%)	41 (43.2%)	12 (37.5%)	
Moderate	9 (28.1%)	35 (36.8%)	13 (40.6%)	
Severe	8 (25.0%)	9 (9.5%)	4 (12.5%)	
Postoperative Outcomes				
EF	38.6 (15.5)	58.2 (11.4)	39.7 (13.7)	<0.001
MR grade (*n*%)				0.112
None	3 (9.4%)	7 (7.3%)	3 (9.4%)	
Mild	16 (50.0%)	23 (24.0%)	13 (40.6%)	
Moderate	10 (31.3%)	53 (55.2%)	14 (43.8%)	
Severe	3 (9.4%)	13 (13.5%)	2 (6.3%)	
Stroke volume	54.9 (19.2)	65.2 (29.3)	70.2 (36.3)	0.098
LVEDD	6.1 (1.8)	5.4 (1.2)	11.5 (19.7)	0.006
LVESD	5.1 (1.9)	3.9 (1.4)	6.5 (6.4)	0.001
LVEDV	136.8 (60.3)	90.6 (44.9)	124.2 (54.5)	<0.001
LVESV	82.9 (50.8)	40.8 (29.5)	69.8 (42.9)	<0.001
Mean gradient (mmHg)	4.8 (2.6)	5.4 (3.5)	4.5 (2.6)	0.262
LAESV	92.9 (33.0)	96.9 (47.3)	98.5 (50.0)	0.883
RVSP	42.2 (10.8)	42.5 (13.2)	47.8 (14.1)	0.123
Tricuspid insufficiency (*n*%)				0.541
None	7 (21.9%)	18 (18.6%)	3 (9.1%)	
Mild	12 (37.5%)	34 (35.1%)	15 (45.5%)	
Moderate	11 (34.4%)	39 (40.2%)	11 (33.3%)	
Severe	2 (6.2%)	6 (6.2%)	4 (12.1%)	
Follow-Up				
EF	38.9 (14.6)	58.8 (10.5)	42.2 (16.3)	<0.001
MR grade (*n*%)				0.181
None	0 (0)	7 (7.4%)	1 (3.1%)	
Mild	10 (31.2%)	19 (20.0%)	9 (28.1%)	
Moderate	19 (59.4%)	43 (45.3%)	15 (46.9%)	
Severe	3 (9.4%)	26 (27.4%)	7 (21.9%)	
Mean gradient (mmHg)	4.8 (2.3)	5.4 (2.6)	4.3 (2.4)	0.087
LVEDD	6.3 (1.6)	5.0 (1.0)	8.9 (18.8)	0.071
LVESD	5.2 (1.9)	3.5 (1.2)	6.2 (10.4)	0.013
Stroke volume	54.5 (25.9)	61.5 (26.6)	56.8 (27.7)	0.387

Note: continuous variables are displayed as means with standard deviations.

**Table 6 jpm-14-00978-t006:** (**A**) Risk factors for all-cause mortality in patients with systolic heart failure. (**B**) Risk factors for all-cause mortality in patients with diastolic heart failure. (**C**) Risk factors for all-cause mortality in patients with mixed heart failure.

(**A**)		
**All-Cause Mortality**	**HR (95% CI)**	***p*-Value**
Dialysis	5.9 (1.1, 30.8)	**0.036**
**Cardiac Death**	SHR (95% CI)	*p*-Value
Mediastinal Radiation	11.8 (2.5, 55.2)	**0.002**
Previous PCI	9.4 (1.3, 69.4)	**0.028**
**MACCEs**	HR (95% CI)	*p*-Value
Prior MI	6.4 (1.6, 25.1)	**0.008**
**MI**	SHR (95% CI)	*p*-Value
Immunocompromised	21.7 (3.6, 131)	**0.001**
(**B**)		
**All-Cause Mortality**	HR (95% CI)	*p*-Value
NYHA > 2	3.0 (1.3, 7.0)	**0.011**
Immunocompromised	3.1 (1.2, 8.0)	**0.02**
EF < 50%	0.96 (0.93, 0.98)	**0.007**
**Cardiac Death**	SHR (95% CI)	*p*-Value
NYHA > 2	11.3 (2.7, 48.1)	**0.001**
Prior Valve Surgery	3.4 (1.2, 10.1)	**0.025**
Immunocompromised	3.9 (1.1, 13.3)	**0.033**
**MACCEs**	HR (95% CI)	*p*-Value
Immunocompromised	4.6 (2.1, 10.0)	**<0.0001**
**Stroke**	SHR (95% CI)	*p*-Value
Prior Valve Surgery	4.1 (1.0, 16.2)	**0.047**
**Repeat Intervention**	SHR (95% CI)	*p*-Value
Immunocompromised	9.4 (1.9, 46.2)	**0.006**
Cardiogenic Shock	13.5 (1.7, 104)	**0.013**
Home O2 use	13.9 (3.1, 61.7)	**0.001**
**New Pacemaker Implantation**	SHR (95% CI)	*p*-Value
Age	0.95 (0.9, 0.99)	**0.009**
Cardiogenic Shock	12.8 (1.9, 85.9)	**0.008**
(**C**)		
**All-Cause Mortality**	HR (95% CI)	*p*-Value
New Pacemaker Implantation	4.7 (1.6, 13.8)	**0.005**
TIA	3.7 (1.03, 13.2)	**0.044**
Pneumonia	3.6 (1.1, 11.5)	**0.028**
COPD	8.0 (1.7, 36.6)	0.008
Diabetes	3.6 (1.2, 11.3)	0.027
**Cardiac Death**	SHR (95% CI)	*p*-Value
Age	0.9 (0.9, 0.99)	0.028
**MAACEs**	HR (95% CI)	*p*-Value
Prior Valve Surgery	2.1 (1.2, 3.9)	0.015
**Stroke**	SHR (95% CI)	*p*-Value
NA		
**MI**	SHR (95% CI)	*p*-Value
Age	1.3 (1.1, 1.5)	0.003

Note: HR: hazard ratio, SHR: sub-distribution hazard ratio (competing risk).

**Table 7 jpm-14-00978-t007:** Cost analysis for the systolic vs. diastolic vs. combined HF groups.

Mean (SD)	Systolic HF *n* = 27	Diastolic HF *n* = 79	Both (Systolic + Diastolic) *n* = 32	*p*-Value
Total Cost	USD 106,859 (USD 52,858)	USD 91,731 (USD 57,177)	USD 120,522 (USD 77,263)	0.08
Direct Cost	USD 65,362 (USD 30,602)	USD 57,094 (USD 35,393)	USD 72,465 (USD 42,259)	0.118
Indirect Cost	USD 41,496 (USD 22,887)	USD 34,637 (USD 22,037)	USD 48,058 (USD 35,391)	0.04
Median (IQR)				
Total Cost	USD 98,890 (60 k–126 k)	USD 71,949 (56 k–114 k)	USD 93,974 (65 k–148 k)	0.127
Direct Cost	USD 61,878 (38 k–80 k)	USD 45,260 (35 k–68 k)	USD 60,170 (41 k–89 k)	0.114
Indirect Cost	USD 37,012 (22 k–54 k)	USD 27,730 (20 k–45 k)	USD 33,803 (24 k–55 k)	0.111

## Data Availability

The data that support the findings of this study are available upon reasonable request to Dr. Serge Sicouri, pending institutional approval.

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
