# Peer review of "Clinical Outcomes and Cost Analysis in Patients with Heart Failure Undergoing Transcatheter Edge-to-Edge Repair for Mitral Valve Regurgitation"

_jpm, 2024, doi:10.3390/jpm14090978_

Round 1

Reviewer 1 Report

Comments and Suggestions for Authors

The manuscript titled "Clinical Outcomes and Cost Analysis in Patients with Heart Failure Undergoing Transcatheter End-to-End Repair for Mitral Valve Regurgitation" lead by research star Dr. Dokollari, evaluates the clinical outcomes and cost implications of transcatheter edge-to-edge repair (TEER) for mitral valve regurgitation (MR) in patients with different types of heart failure (HF): systolic, diastolic, and mixed (systolic + diastolic). The study involves 162 HF patients who underwent TEER (2019-2023). After propensity-adjusted analyses, the clinical outcomes and cost were compared. The propensity-adjusted analysis and the identification of risk predictors are noteworthy strengths and the mansucript was written pretty well. However, the manuscript requires more detailed reporting of the statistical methods, patient characteristics, and clinical outcomes to strengthen its conclusions. Here are my detailed suggestions:

Major Comments:

1.      Statistical Analysis and Reporting:

    1. The manuscript predominantly reports results using mean and standard deviation (SD), but switches to medians in some outcome tables. Continuous variables in cardiac surgery, especially in small samples, are rarely normally distributed. Please check for variable normality and clarify which tests were used to assess this. For skewed distributions, report the data as median (Q1, Q3) and use nonparametric tests for comparisons. For normally distributed data, parametric tests and mean (SD) reporting are appropriate.
    2. The propensity-adjusted analyses should be thoroughly described. The manuscript should provide a more detailed explanation of the matching process and the specific covariates used.
    3. The statement "As shown by multiple studies, propensity-adjustments provide similar or better adjustment" may not hold true in all cases. Methods like inverse probability weighting (IPW) can be superior depending on the scenario. If you are citing multiple studies to support this claim, please include references to comparative studies showing superiority in similar contexts. Alternatively, consider revising the language to acknowledge that while propensity adjustments are widely used, the choice of method should be context-dependent.

2.      Clinical Outcomes and Follow-Up:

1.      The manuscript should explore potential reasons for the observed differences in clinical outcomes among the HF groups, considering both physiological and procedural factors.

2.      The manuscript mentions follow-up data being collected via outpatient clinics and referring cardiologists. However, it does not detail the completion rate of this follow-up or how non-responses from referring cardiologists were handled. This could introduce bias, particularly if non-response is correlated with worse outcomes. Please include a detailed analysis of follow-up completion rates and the potential impact of missing data on the study’s conclusions.

3.      Cost Analysis:

1.      The cost analysis period (January 2018 to September 2022) is shorter than the overall study period (March 2017 to September 2022). Please clarify the rationale for this discrepancy. It is essential to explain why the cost data was limited to this timeframe and how this might impact the generalizability of the cost analysis results.

2.      Cost Calculation Method: Describe the methodology used to calculate costs, including any adjustments made for inflation or regional cost variations. Explain how costs were aggregated and whether any cost categories were excluded from the analysis.

3.      The manuscript should discuss the implications of the observed cost differences (or lack thereof) on clinical practice and healthcare policy.

4.      Consider conducting a cost-effectiveness analysis if feasible, or discussion in future research to compare the costs and clinical outcomes of TEER in different HF groups to assess the value of the intervention. Discuss the potential cost savings or increased expenditures associated with TEER in each group and the implications for healthcare decision-making.

4.      Study Limitations:

1.      Retrospective Nature: Address the limitations inherent to the retrospective study design, such as potential selection bias and unmeasured confounding. Acknowledge the limitations of the propensity-adjusted analysis in fully eliminating these biases.

2.      Single-Center Study: Discuss the generalizability of the findings given the single-center nature of the study. Consider how institutional practices and patient demographics, as well as hospital-based charges might differ from other settings.

Minor Comments:

5.      Tables:

 In the tables, combining beta coefficients (B) and odds ratios (OR) in the same column could be confusing for readers. I suggest either separating them into different columns or indicating which metric is used for each variable by annotating the variable names accordingly (in parenthesis).

6.      Figures:

Kaplan-Meier survival curves should include confidence intervals (CI) and the number of patients at risk at various time points. The confidence intervals are best displayed as shaded areas or bars in the figure, and the number of patients at risk should be shown as a table below the graph. This will provide a more complete and informative presentation of the survival data.

  1. To support open-source and full reproducibility, I suggest that the authors produce their DO files for readers. This will allow readers to follow a step-by-step approach to apply the analysis methods and allow us to look at the histograms of normally distributed data.

Author Response

Major Comments:

Statistical Analysis and Reporting:

  1. The manuscript predominantly reports results using mean and standard deviation (SD), but switches to medians in some outcome tables. Continuous variables in cardiac surgery, especially in small samples, are rarely normally distributed. Please check for variable normality and clarify which tests were used to assess this. For skewed distributions, report the data as median (Q1, Q3) and use nonparametric tests for comparisons. For normally distributed data, parametric tests and mean (SD) reporting are appropriate.
  • We thank the reviewer for this comment. We did test normality and have updated the statistical analysis section to reflect that. In particular, the values in Table 5 are displayed as means with standard deviations because the means were similar to the medians, and we determined that means are easier to interpret.

2. The propensity-adjusted analyses should be thoroughly described. The manuscript should provide a more detailed explanation of the matching process and the specific covariates used.

  • We thank the reviewer for this suggestion. We added additional explanations to a supplementary document. Please note that no matching was done in this analysis. We used the propensity score as a covariate in the models.

3. The statement "As shown by multiple studies, propensity-adjustments provide similar or better adjustment" may not hold true in all cases. Methods like inverse probability weighting (IPW) can be superior depending on the scenario. If you are citing multiple studies to support this claim, please include references to comparative studies showing superiority in similar contexts. Alternatively, consider revising the language to acknowledge that while propensity adjustments are widely used, the choice of method should be context-dependent.

  • We thank the reviewer for this suggestion. We removed the sentence that claims propensity adjustments provide similar or better adjustment.

Clinical Outcomes and Follow-Up:

  1. The manuscript should explore potential reasons for the observed differences in clinical outcomes among the HF groups, considering both physiological and procedural factors.
  • We thank the reviewer for this comment. In response, we have expanded the discussion section to include a more comprehensive analysis of the variations in physiological and procedural factors observed in both groups.

  1. The manuscript mentions follow-up data being collected via outpatient clinics and referring cardiologists. However, it does not detail the completion rate of this follow-up or how non-responses from referring cardiologists were handled. This could introduce bias, particularly if non-response is correlated with worse outcomes. Please include a detailed analysis of follow-up completion rates and the potential impact of missing data on the study’s conclusions.
  • We thank the reviewer for this comment. The follow-up completion rate was 97%, we included this sentence in the text.

Cost Analysis:

  1. The cost analysis period (January 2018 to September 2022) is shorter than the overall study period (March 2017 to September 2022). Please clarify the rationale for this discrepancy. It is essential to explain why the cost data was limited to this timeframe and how this might impact the generalizability of the cost analysis results.
  • We thank you the reviewer for giving us the opportunity to explain this point. Data on the cost of hospital stays was only available for Jan 2018 to Sept 2022 because of a change in our electronic medical record system.

  1. Cost Calculation Method: Describe the methodology used to calculate costs, including any adjustments made for inflation or regional cost variations. Explain how costs were aggregated and whether any cost categories were excluded from the analysis.
  • In the cost analysis section of the methods, we include an exhaustive list of what was included in the cost calculation. We did not adjust for inflation.

  1. The manuscript should discuss the implications of the observed cost differences (or lack thereof) on clinical practice and healthcare policy.
  • We appreciate the reviewer for allowing us to further discuss this aspect. Our study shows no significant cost differences among the groups. We address this implication in the discussion section on cost analysis.

  1. Consider conducting a cost-effectiveness analysis if feasible, or discussion in future research to compare the costs and clinical outcomes of TEER in different HF groups to assess the value of the intervention. Discuss the potential cost savings or increased expenditures associated with TEER in each group and the implications for healthcare decision-making.
  • Dear reviewer, a cost-effectiveness study does not seem appropriate for this study because we are not comparing different treatments.

Study Limitations:

  1. Retrospective Nature:Address the limitations inherent to the retrospective study design, such as potential selection bias and unmeasured confounding. Acknowledge the limitations of the propensity-adjusted analysis in fully eliminating these biases.
  2. Single-Center Study:Discuss the generalizability of the findings given the single-center nature of the study. Consider how institutional practices and patient demographics, as well as hospital-based charges might differ from other settings.
  • We appreciate the valuable suggestion from the reviewer. As per the recommendation, we have included a limitations section covering the two suggested points.

Minor Comments:

Tables:

  1. In the tables, combining beta coefficients (B) and odds ratios (OR) in the same column could be confusing for readers. I suggest either separating them into different columns or indicating which metric is used for each variable by annotating the variable names accordingly (in parenthesis).
  • We appreciate the reviewer for this suggestion. We added a note to the bottom of the table. Adding more columns will make the table too wide.

Figures:

  1. Kaplan-Meier survival curves should include confidence intervals (CI) and the number of patients at risk at various time points. The confidence intervals are best displayed as shaded areas or bars in the figure, and the number of patients at risk should be shown as a table below the graph. This will provide a more complete and informative presentation of the survival data.
  • We thank the reviewer for this comment. The figures shown are not Kaplan-Meier survival curves, we apologize for the error in labeling. They are survival and cumulative incidence curves created after the propensity-adjusted Cox and Fines and Grays regression models. These graphs do not include the number of patients at risk or confidence intervals. Please see this Stata manual for more info: pdf (stata.com)

To support open-source and full reproducibility, I suggest that the authors produce their DO files for readers. This will allow readers to follow a step-by-step approach to apply the analysis methods and allow us to look at the histograms of normally distributed data.

  • We thank you the reviewer for this suggestion. We will consider creating DO files, if possible.

Reviewer 2 Report

Comments and Suggestions for Authors

The article aims to investigate the outcomes of patients with various types of heart failure undergoing TEER procedures and to also conduct a cost analysis among different groups. I congratulate the authors because the objective of the study is very interesting and ambitious. However, I must admit that the study cannot be accepted in its current form and should be revised in many aspects. I would also like to emphasize that the article is not easy to read and is structured with a lot of tabular material that is not adequately explained in the text.

Below, I provide detailed comments on the manuscript:

- The outcomes of the study are not clearly defined in the abstract.

- abstract: Ventilation time is lower in the diastolic group, not higher (2 vs 2.3).

- The acronym MACCE should be defined in the methods section of the abstract.

- The guidelines, including the 2022 AHA guidelines referenced by the authors, classify heart failure as HFrEF, HFmrEF, and HFpEF. However, the authors have defined the patients as having systolic and diastolic heart failure. I believe the authors should change the group names to patients with HFpEF, HFmrEF, and HFrEF (if this was the intended division). The terms systolic or diastolic heart failure or mixed are obsolete and are not included in the guidelines referenced by the authors anyway.

- The T-Test and the sum rank test (reported in the statical methods) are not applicable to this study because the groups are more than two. The authors probably used the ANOVA test and the Kruskal-Wallis test with a post-hoc correction for the differences between groups. Please review this crucial point.

- Regarding MACCE, the authors refer to "repeat intervention." It is not clear what this refers to. Do they mean the repetition of the TEER procedure? In any case, this should be specified more clearly.

- Why did the authors decide to include myocardial infarction as an outcome within the composite outcome? Myocardial infarction does not seem to be related to post-procedural outcomes of patients undergoing TEER.

- How is it possible that in the systolic HF group (or HFrEF), only 68.8% of patients have EF <50%? By definition, patients with systolic heart failure or HFrEF should all have EF <50%.

- I do not understand why the authors report an HR when comparing groups with continuous or categorical variables. For example: "diastolic HF patients had a higher total ventilation hours [HR 49.4 (8.6, 90.2)]." Usually, this type of difference should be expressed by means or medians or by percentages, and only the P value of the test should be reported (i.e., T-Test or Chi-squared test). HR usually refers to Cox regression analysis and does not represent differences between group variables but differences in survival between groups regarding a specified outcome. Please report differences between groups as appropriate.

- The tables are all too long; some should be divided into multiple tables. Additionally, there are too many variables reported! Most of the variables reported are irrelevant to the study and should be eliminated as they only serve to heavily burden the manuscript (i.e., endocarditis with 0 cases in all groups???, report only males, not both male and females).

- In Table 2, there is a P value that is not clear if it refers to the analysis with PS or without PS.

- Furthermore, I do not understand what is meant by OR in the PS adjusted analysis. What is the outcome considered for this OR?

- The tables do not have captions.

- Use the name of the molecule and not the commercial name (clopidogrel instead of Plavix).

- Honestly, I do not understand what Table 4 represents at all.

- I did not understand how the univariate and multivariate Cox regression were performed. Generally, this type of analysis is used to assess whether different variables of a single sample have a predictive value on an outcome over time (i.e.: EF predicts death independently from age after AMI, where age and EF are variables of the same group). Here, univariate and multivariate analyses compare the state of systolic, diastolic, and mixed heart failure, which are in fact three different groups. Usually, survival differences between three different groups are studied with Kaplan-Meier and the log-rank test.

Author Response

We thank the reviewer very much for the time to review this manuscript.

  1. The outcomes of the study are not clearly defined in the abstract.
  • Based on the reviewer’s suggestion, we have included the primary outcomes in the abstract.

  1. abstract: Ventilation time is lower in the diastolic group, not higher (2 vs 2.3).
  • We thank the reviewer for the suggestion. We have made the change in the text.

  1. The acronym MACCE should be defined in the methods section of the abstract.
  • We greatly appreciate the reviewer's suggestion. We have clearly defined MACCE in the abstract.

  1. The guidelines, including the 2022 AHA guidelines referenced by the authors, classify heart failure as HFrEF, HFmrEF, and HFpEF. However, the authors have defined the patients as having systolic and diastolic heart failure. I believe the authors should change the group names to patients with HFpEF, HFmrEF, and HFrEF (if this was the intended division). The terms systolic or diastolic heart failure or mixed are obsolete and are not included in the guidelines referenced by the authors anyway.
  • We thank the reviewer for this suggestion. As per AHA guidelines, we have renamed systolic HF as HFrEF, diastolic HF as HFpEF, and combined HF as HFmrEF in our paper.

  1. The T-Test and the sum rank test (reported in the statical methods) are not applicable to this study because the groups are more than two. The authors probably used the ANOVA test and the Kruskal-Wallis test with a post-hoc correction for the differences between groups. Please review this crucial point.
  • "We thank the reviewer and greatly appreciate this comment. After carefully reviewing the statistical methods section, we have incorporated the following compelling paragraph: “Descriptive statistics we used for all pre, intra, and post-operative variables. Initial comparison of pre, intra, post-operative, and echocardiographic variables by heart failure groups were done with one-way Anova or Kruskal-Wallis for continuous variables and chi-square for categorical variables. Three propensity scores were created using multivariable logistic regression with the groups systolic HF vs diastolic HF, systolic HF vs mixed HF, and diastolic vs mixed HF as the dependent variables for each score. Pre-operative variables that differed between the groups were entered into the models as independent variables. Intra and post-operative outcomes were compared with propensity-adjusted regression models with beta coefficients displayed for continuous variables, odds ratios for categorical variables, and 95% confidence intervals. Long-term outcomes of the HF groups were compared with cumulative incidence per 100 person-years and log-rank tests; the number of risk and frequency and percentage of events at 1, 2, and 5 years; and finally univariable, propensity-adjusted, and multivariable Cox regression and Fines and Grays analyses. Kaplan Meier figures were created to compare the long-term outcomes by HF groups with a log-rank test. In addition, cubic spline graphs using a GLM with poisson family, log link, and robust estimates were produced to visualize changes in risk ratio over time. Finally, risk factors for the long-term outcomes by HF group were found by using a forward selection with high p-value criteria (0.5) and AIC. The risk factor analysis used Cox regression for all-cause mortality and MACCE and Fines and Grays competing risk regression for all other outcomes. 

  1. Regarding MACCE, the authors refer to "repeat intervention." It is not clear what this refers to. Do they mean the repetition of the TEER procedure? In any case, this should be specified more clearly.
  • We appreciate the reviewer for the opportunity to clarify this point further. We refer to repeat intervention whether transcatheter or through open-heart surgery. We have added it to the text.

  1. Why did the authors decide to include myocardial infarction as an outcome within the composite outcome? Myocardial infarction does not seem to be related to post-procedural outcomes of patients undergoing TEER.
  • We appreciate the reviewer's comment. Myocardial infarction following TEER is rare but has been reported previously, with an incidence of 0.7% (Maisano F, Franzen O, Baldus S, et al. Percutaneous mitral valve interventions in the real world: early and 1-year results from the ACCESS-EU, a prospective, multicenter, nonrandomized post- approval study of the MitraClip therapy in Europe. J Am Coll Car- diol. 2013; 62(12): 1052–1061) and 0,1% (Sorajja P, Vemulapalli S, Feldman T, et al. Outcomes with tran-scatheter mitral valve repair in the United States: an STS/ACC TVT registry report. J Am Coll Cardiol. 2017; 70(19): 2315–2327). Furthermore, this is part of the MACCE definition (composite of stroke, systemic embolism, myocardial infarction, unstable angina requiring revascularization, or death from any cause). Therefore, we consider it a relevant parameter in our study.

  1. How is it possible that in the systolic HF group (or HFrEF), only 68.8% of patients have EF <50%? By definition, patients with systolic heart failure or HFrEF should all have EF <50%.
  • We thank you the reviewer for this comment. As per AHA guidelines, HFrEF is HF with LVEF ≤40%, HFmrEF is symptomatic HF with LVEF 41-49% and HFpEF is symptomatic HF with LVEF ≥50%. Heidenreich PA, Bozkurt B, Aguilar D, et al. 2022 AHA/ACC/HFSA guideline for the management of heart failure. J Am Coll Cardiol2022; 79: e263–e421.

  1. I do not understand why the authors report an HR when comparing groups with continuous or categorical variables. For example: "diastolic HF patients had a higher total ventilation hours [HR 49.4 (8.6, 90.2)]." Usually, this type of difference should be expressed by means or medians or by percentages, and only the P value of the test should be reported (i.e., T-Test or Chi-squared test). HR usually refers to Cox regression analysis and does not represent differences between group variables but differences in survival between groups regarding a specified outcome. Please report differences between groups as appropriate.
  • Dear reviewer, hazard ratio (HR) is a measure of the effect of an intervention in an outcome of interest over time. HR is reported most commonly in time-to-event analysis or survival analysis (i.e. when we are interested in knowing how long it takes for a particular event/outcome to occur). In our case, with the propensity-adjusted analysis, we can determine if there are any differences. Then with HR, we aim to evaluate at what point in time the outcomes occur.

  1. The tables are all too long; some should be divided into multiple tables. Additionally, there are too many variables reported! Most of the variables reported are irrelevant to the study and should be eliminated as they only serve to heavily burden the manuscript (i.e., endocarditis with 0 cases in all groups???, report only males, not both male and females).
  • We appreciate this comment by the reviewer. However, Table 1 represents baseline characteristics used to create the propensity score and determine risk factors for long-term outcomes. Removing these variables would make it difficult for future researchers to replicate this study. We can shorten some of the variables as you suggested like only report males and take out endocarditis.

  1. In Table 2, there is a P value that is not clear if it refers to the analysis with PS or without PS. Furthermore, I do not understand what is meant by OR in the PS-adjusted analysis. What is the outcome considered for this OR?
  • We appreciate the reviewer for providing this comment. The p-values for Table 2 are from ANOVA and chi-square tests. The propensity-adjusted results have 95% confidence intervals instead of p-values. We can add a note to the bottom of Table 2 for more clarity. In addition, OR = odds ratio, again we will add a note for clarity.

  1. The tables do not have captions. We thank the reviewer for this suggestion. We have added captions to the tables.

  1. Use the name of the molecule and not the commercial name (clopidogrel instead of Plavix).
  • Based on the reviewer's suggestion, we have included the name of the molecule in the text.

  1. Honestly, I do not understand what Table 4 represents at all.
  • Dear reviewer, Table 4 represents a simple analysis of long-term outcomes. We have looked at the number at risk (number still in the study) at 1, 2, and 5 years. In addition, the columns to the right display the frequency and percentages of the events at each time point.

  1. I did not understand how the univariate and multivariate Cox regression were performed. Generally, this type of analysis is used to assess whether different variables of a single sample have a predictive value on an outcome over time (i.e.: EF predicts death independently from age after AMI, where age and EF are variables of the same group). Here, univariate and multivariate analyses compare the state of systolic, diastolic, and mixed heart failure, which are in fact three different groups. Usually, survival differences between three different groups are studied with Kaplan-Meier and the log-rank test.
  • We appreciate the reviewer for allowing us to provide additional clarification on this point. Cox regression and Fines and Gray regression are what you describe, a way to assess the predictive value of variables with an outcome over time. Our sample is one sample, not three separate samples. We have one dataset with a categorical variable to designate whether a person had systolic, diastolic, or mixed HF. You describe a continuous variable (EF) being associated with death independently from age. Similar models can be built using categorical variables with one category acting as the reference variable. We added the propensity score to the propensity-adjusted models in addition to the HF variable. For multivariable models, we controlled for potential confounders. These models show which HF groups predict the long-term outcomes.  For more info, see this link: https://doi.org/10.1155%2F2021%2F1302811

Reviewer 3 Report

Comments and Suggestions for Authors

The paper and the topic is important but the whole presentation of the results is confusing.

Major concerns on the large number of the tables of the paper. 

Furthermore the cost is the one of the main purpose of this study but it's presenting shortly

The discussion is poor and should be enriched.

Comments on the Quality of English Language

The whole presentation is adequate.

Author Response

We thank the reviewer for taking the time to review our paper. The results are presented in a sequence that follows the research questions and objectives outlined earlier in the manuscript. Each section builds on the previous one, allowing the reader to follow the progression of the analysis. The results are divided into distinct subsections, each addressing specific aspects of the research. This segmentation helps to avoid overwhelming the reader with too much information at once and makes it easier to digest the findings. Each section is also followed by the corresponding Table and/or Figure, as per author guidelines. I agree that this could be confusing. If there are specific sections or elements that you found confusing, we would appreciate further details so that we can address those directly.

The large number of tables ensures that all relevant aspects of the data are presented, leaving no stone unturned. This thoroughness adds to the credibility of the research, as it demonstrates that every possible angle has been considered and analyzed. In addition, we believe that tables serve as a transparent record of the data collected and the analysis performed. This transparency is crucial for the reproducibility of the research, allowing other researchers to replicate the study or build upon it. However, to improve readability, we have relocated Table 6 to the supplementary material.

We have enhanced both the discussion and cost analysis sections based on your suggestion.

Round 2

Reviewer 2 Report

Comments and Suggestions for Authors

We would like to express our sincere gratitude to the authors for their thorough and thoughtful responses to the comments provided. Their engagement with the feedback has markedly enhanced the quality of the manuscript, demonstrating a commendable commitment to refining their work. The revisions made not only address the previous concerns but also contribute to a clearer presentation of their research objectives and findings. 

In addition to the improvements noted, we would like to offer a suggestion regarding the manuscript's title. We recommend replacing the term “end to end” with “edge to edge.” This substitution better encapsulates the focus and scope of the study, aligning with contemporary terminology and standards in the relevant field.

Author Response

We are grateful to the reviewer for the suggestion. We have agreed to modify the title of the manuscript in line with the feedback.

Reviewer 3 Report

Comments and Suggestions for Authors

I think that is a interesting study focused on a very hot topic that of TEER in patients with heart failure and severe mitral regurgitation. It's known that this group of patients had poor prognosis with several hospitalizations.

The authors presented the outcome and the costs of the whole patient management in this very difficult group of patients.

The paper has many tables and is enriched with many data.

The paper is long extended. 

Author Response

Dear reviewer, 

thank you for your comments and suggestions. 

We appreciate the time and effort you have dedicated to improving the quality of our manuscript. As mentioned previously, we have conducted a thorough analysis in order to provide comprehensive answers to the readers' questions and hopefully provide them with valuable insights. We would also appreciate your input and suggestions on what you believe should be specifically added or trimmed in the manuscript to benefit the readers.

thank you